# Apical Sealing Ability of Two Calcium Silicate-Based Sealers Using a Radioactive Isotope Method: An In Vitro Apexification Model

**DOI:** 10.3390/ma14216456

**Published:** 2021-10-27

**Authors:** Inês Raquel Pereira, Catarina Carvalho, Siri Paulo, José Pedro Martinho, Ana Sofia Coelho, Anabela Baptista Paula, Carlos Miguel Marto, Eunice Carrilho, Maria Filomena Botelho, Ana Margarida Abrantes, Manuel Marques Ferreira

**Affiliations:** 1Faculty of Medicine, Institute of Endodontics, University of Coimbra, 3000-075 Coimbra, Portugal; inespereira98@gmail.com (I.R.P.); craveiro_catarina@hotmail.com (C.C.); sirivpaulo@gmail.com (S.P.); josepedromartinho@gmail.com (J.P.M.); 2Center for Innovative Biomedicine and Biotechnology (CIBB), University of Coimbra, 3000-548 Coimbra, Portugal; anasofiacoelho@gmail.com (A.S.C.); anabelabppaula@sapo.pt (A.B.P.); cmiguel.marto@uc.pt (C.M.M.); eunicecarrilho@gmail.com (E.C.); mfbotelho@fmed.uc.pt (M.F.B.); mabrantes@fmed.uc.pt (A.M.A.); 3Clinical Academic Center of Coimbra (CACC), 3004-561 Coimbra, Portugal; 4Coimbra Institute for Clinical and Biomedical Research (iCBR) Area of Environment, Genetics and Oncobiology (CIMAGO), University of Coimbra, 3000-548 Coimbra, Portugal; 5Faculty of Medicine, Institute of Integrated Clinical Practice, University of Coimbra, 3000-075 Coimbra, Portugal; 6Faculty of Medicine, Institute of Experimental Pathology, University of Coimbra, 3000-548 Coimbra, Portugal; 7Faculty of Medicine, Institute of Biophysics, University of Coimbra, 3000-548 Coimbra, Portugal

**Keywords:** apexification, calcium silicate, dental leakage, mineral trioxide aggregate, sodium pertechnetate Tc 99m

## Abstract

The aim of this study was to evaluate and compare the sealing ability of two calcium silicate-based sealers (TotalFill BC RRM Fast Set Putty and White ProRoot MTA) when used as apical plugs in immature teeth through nuclear medicine. Single-rooted extracted teeth (n = 34) had their crowns and root tip sectioned to obtain 14 mm long root segments to simulate an in vitro apexification model. Were created two experimental groups, namely MTA (n = 12) and BC (n = 12), and two control groups, PG (positive group, n = 5) and NG (negative group, n = 5). On the 4th day after placing the respective apical plug, the apical portions of the teeth were submerged in a solution of sodium pertechnetate (^99m^TcNaO_4_) for 3 h. Statistical analysis showed a significant difference between the MTA group and the controls (*p* < 0.05). The BC group had a significant difference regarding the negative control (*p* < 0.001) but showed no statistical significance regarding the positive control (*p* = 0.168). There was a statistically significant difference (*p* = 0.009) between the BC group (7335.8 ± 2755.5) and the MTA group (4059.1 ± 1231.1), where the last showed less infiltration. Within the limitations of this study, White ProRoot MTA had a significantly better sealing ability than TotalFill BC RRM Fast Set Putty.

## 1. Introduction

Pulp necrosis in immature teeth can be caused by trauma, damaging Hertwig’s root sheath (HERS) [1], or by late diagnosis and treatment of extensive carious lesions [2]. The complete development of such teeth is interrupted, which results in a compromised root formation that leads to an unreached final root length; thin root dentinal walls that are related to the presence of a wide canal space and open apex and also resulting in an apical divergence causing the lack of an apical constriction. [3]. Such inherent characteristics can be an endodontic challenge [4]. Essentially the divergent apical architecture and absence of apical constriction are the main factors of irrigants, intracanal dressings, and root canal fillings extrusion into the periapical soft tissues, making the complete debridement and control of the filling nearly impossible, compromising the long-term outcome of the treatment [4,5,6].

These challenges require specific approaches such as a chemical disinfection protocol with minimal instrumentation and a softened filling technique [7,8]. The current treatment choices for immature teeth are apexification, placing an apical plug, or regenerative endodontic treatment [8].

The traditional approach to handle cases of necrotic teeth with an open apex used to be a multiple-visit apexification treatment using calcium hydroxide (Ca(OH)_2_) as intracanal dressing [9]. Although Ca(OH)_2_ apexification has high clinical success rates and most of the teeth exhibit a calcified barrier at the apical portion. Several limitations led to seek for alternative treatment modalities [9]. The long time for the apical barrier to be formed, the need for multiple visits, the possibility of coronal microleakage as well as the lower resistance to fracture of the teeth during treatment are some of those disadvantages [9].

The placement of an apical plug is one alternative to calcium hydroxide apexification [10]. Apical plug is described as non-surgical compaction of biocompatible material into the apical portion of the root canal that prevents the infiltration of toxins and bacteria into periradicular tissues [11,12]. This barrier enables the placement of an appropriate root canal sealant and filling material while reducing the possibility of their extrusion into periapical tissues, preventing overfilling and promoting periapical healing [7,12,13].

A proper apical sealing ability has been proved to be the most important characteristic for achieving success in endodontic procedures because it is responsible for preventing the passage of bacteria and their byproducts from the oral environment into the periapical tissues [5,14]. Several studies have demonstrated that an apical plug of 3 to 5 mm is sufficient to achieve an adequate seal in apical barrier techniques [5,15,16].

Therefore, an ideal root canal sealer should seal the pathway between the root canal space and the surrounding tissues, providing resistance to leakage [17,18]. In addition, it needs to be nontoxic, radiopaque, biocompatible, bacteriostatic, insoluble in tissue fluids, dimensionally stable, easy to prepare and handle and have a proper setting time and biomimetic properties [17,18,19]. Although none meets all properties of Grossman’s ideal, there are many root canal sealers available that can be grouped into their main chemical components: zinc oxide eugenol, calcium hydroxide, glass ionomer, silicone, resin, and bioceramic-based cements [20,21,22].

Bioceramic-based sealers are relatively recent in endodontic history, being used for the past 30 years and designed specifically for medical and dental use [20]. They are biocompatible and, therefore, nontoxic and chemically stable within the biological environment [23]. The main advantage of this sealer is its ability to form hydroxyapatite to create a bond with the dentin structure, promoting its sealing ability [23,24]. Bioceramics are classified into bioactive or bioinert materials according to their behavior within the surrounding living tissue [20]. Bioactive materials, such as glass and calcium phosphate, interact to promote the growth of more durable tissues [20]. Bioinert materials, such as zirconia and alumina, produce an insignificant response, having no biological or physiological effect [20]. Bioactive materials are additionally classified according to their stability as degradable or nondegradable [20].

Apical plug with mineral trioxide aggregate (MTA) has become the gold standard for endodontic treatment of immature teeth [5,19,25]. It is used for root-end filling treatments and also for procedures such as pulp capping, pulpotomy, apexogenesis, apexification, root perforations repair, and root canal filling [19,25]. MTA is a calcium-silicate-based sealer with dicalcium silicate, tricalcium silicate, tricalcium aluminate, bismuth oxide, and calcium sulfate in its composition [26,27,28]. However, MTA has some disadvantages, including tooth discoloration, reports of challenging manipulation, prolonged setting time, and high cost [5,17,29]. White MTA (WMTA) is now the most adopted choice to obtain better aesthetic results due to reports of coronal discoloration with gray MTA (GMTA) [1,17,30].

As a result of these drawbacks, the investigation for an ideal apical plug material is still in progress [29].

Recently, new bioceramic materials have been launched as an alternative to MTA, which owns many similar and some different characteristics, in an effort to develop the Grossman’s ideal material [23] and with claimed improved handling characteristics [23,27,31]. They are composed of tricalcium silicate, dicalcium silicate, zirconium oxide, tantalum pentoxide, calcium phosphate monobasic (BC), and calcium phosphate [26,27,32]. It can be presented in various formats: paste, putty, and fast set putty [27]. Studies have shown that it does not induce discoloration in the tooth structure as well [32].

However, few published studies have compared the sealing ability of these bioceramic sealers when used as apical plugs in immature teeth.

The aim of the present study is to evaluate and compare the sealing ability of two calcium silicate-based sealers (TotalFill Bioceramic Root Repair Material Fast Set Putty and White ProRoot MTA) used as apical plugs in immature teeth through nuclear medicine.

The null hypothesis is that there are no significant differences in the sealing ability of the calcium silicate-based sealers White ProRoot MTA and the TotalFill BC RRM Fast Set Putty.

## 2. Materials and Methods

### 2.1. Collection of the Sample

For this in vitro study, all the experimental process was performed and observed by the authors and supervised by an endodontic expert. Thirty-four maxillary and mandibular monoradicular premolars that were extracted due to orthodontic or periodontal reasons were used. After that, the periodontal ligament or calculus was removed with Gracey curettes. The teeth were cleaned with NaOCl at 2.5% and preserved in chloramine T at 4 °C until used to prevent bacterial growth.

The teeth selected had to be straight, with a single and permeable canal and fully formed roots. The exclusion criteria were calcified canals, teeth with obstructions inside the canal, internal or external root resorption, previous endodontic treatment, fractures or root fissures, caries lesions, or restorations from cementoenamel junction (CEJ) level to the apex and without pronounced apical curvature.

To standardize the selection of teeth to similar internal anatomy and to confirm the existence of a single canal, preoperative radiographs were performed in orthogonal and proximal projections. The teeth were examined through an optical microscope (Leica^®^ M320, Wetzlar, Germany) to assess that there were no fissures or fractures.

### 2.2. Preparation

To standardize the sample, the crowns were sectioned at the CEJ level with a cylindrical diamond drill (Coltène/Whaledent AG, Altstätten, Switzerland), mounted on a high-speed handpiece, perpendicular to the long root axis, in order to obtain root segments 17 mm long. In the teeth where the canal was not exposed, access was made with a spherical diamond drill (Coltène/Whaledent AG, Altstätten, Switzerland) mounted on a high-speed handpiece.

The apical portion of the roots (3 mm) was sectioned with a cylindrical diamond drill (Coltène/Whaledent AG, Altstätten, Switzerland) mounted on a high-speed handpiece to simulate an immature tooth. Then, the coronal and root surfaces were smoothened with a carborundum disk (Dentorium Products Co. Inc., Farmingdale, NY, USA). A final length of 14 mm was obtained for the specimens.

After that, an exploration and permeabilization of the root canal was made with a 25 mm manual K file, ISO size #10 (Dentsply Maillefer, Ballaigues, Switzerland). The working length was established at 14 mm by inserting a K file ISO #10 (Dentsply Maillefer, Ballaigues, Switzerland) until it was visible in the apical foramen. The biomechanical orthograde preparation was firstly made by manual instrumentation with a manual K file ISO #10, #15, and #20 (Dentsply Maillefer, Ballaigues, Switzerland) and then by mechanical instrumentation with 25 mm Protaper Next (Dentsply Maillefer, Ballaigues, Switzerland) in X1 (17.04), X2 (25.06) and X3 (30.07) sequence at 14 mm, with a constant speed of 250 rpm and torque control of 1.2 N/cm, using the electric motor X-SMART^TM^ (Dentsply Maillefer, Ballaigues, Switzerland). The irrigation was made with 2 mL of 2.5% NaOCl (sodium hypochlorite) with an irrigation needle 27 G, with a close end and lateral opening, between files.

To simulate teeth with open apex (Figure 1), a divergent retrograde preparation was created by ProFile Orifice Shaper (Dentsply Maillefer, Ballaigues, Switzerland) at OS#1 (06/20), OS#2 (06/25), and OS#3 (06/30) sequence introduced at the apical 10 mm, through the electric motor X-SMART^TM^ (Dentsply Maillefer, Ballaigues, Switzerland), with a constant speed of 250 rpm and torque control of 1.2 N/cm. The irrigation procedure was the same as in the orthograde preparation.

A final irrigation was performed with 1 mL of 17% ethylenediaminetetraacetic acid (EDTA) (Magnum Dental AS, Tartu, Estonia) for 1 min (to dissolve the inorganic component of the smear layer), 2 mL of 2.5% NaOCl (to remove the organic component of the smear layer) and 2 mL of 0.9% NaCl. The canal was dried with paper points, and the integrity of the roots was checked with an optical microscope (Leica^®^ M320, Wetzlar, Germany).

### 2.3. Groups

Following the canal preparation, a random distribution of the 34 root segments was performed into 4 groups.

Two experimental groups with 12 teeth each:ProRoot White MTA group (MTAG, n = 12);TotalFill BC RRM Fast Set Putty group (BCG, n = 12).

Two control groups with 5 teeth each:Negative control group (NG, n = 5);Positive control group (PG, n = 5).

### 2.4. Endodontic Sealers

The root canal sealers used were calcium-silicate-based TotalFill BC RRM Fast Set Putty (FKG, La Chaux-de-Fonds, Switzerland) and ProRoot WMTA (Dentsply Maillefer, Ballaigues, Switzerland). The manufacturer, composition, and lot no. are presented in Table 1.

### 2.5. Obturation

A 4 mm apical plug was placed in the 3 groups, except in the PG group, where the teeth were left without apical obturation material. In the NG and the MTAG group, a 4 mm ProRoot White MTA apical plug was placed. In the BCG group, a 4 mm TotalFill BC RRM Fast Set Putty apical plug was placed.

The WMTA was handled according to the manufacturers’ instructions. Regarding the BC RRM Fast Set Putty, as it is a premixed material ready to use, no preparation of the material was necessary. The respective materials were applied with a MAP System (Micro Apical Placement System) (Dentsply Maillefer, Ballaigues, Switzerland) with a diameter of 1.1 mm. The cement was compacted with Schilder Pluggers (Dentsply Maillefer, Ballaigues, Switzerland) with the initial stop marking the 14 mm length, performing increments of 1 mm until the final stop marked a 10 mm length, creating a 4 mm thick apical plug (Figure 2). The placement of the endodontic sealer was performed under an optical microscope (Leica^®^ M320, Wetzlar, Germany).

Each group was placed in its recipient, containing florist sponges previously soaked in chloramine T, to simulate the soft periapical tissues. To create an atmosphere comparable to in vivo, the specimens were stored at room temperature in an environment with 100% relative humidity for 4 days, allowing the complete set of the sealers.

### 2.6. Infiltration

In the experimental and positive control groups, two layers of nail varnish were applied to the outer surface of each root, except at the last 1 mm root tip. In the negative control group, the whole root segment, including the root tip, was sealed with two layers of varnish.

On the 4th day after the placement of the sealers, the apical region of the specimen was submerged in a 50 µL solution (8 mCi/mL) of sodium pertechnetate (^99m^TcNaO_4_) placed in radioimmunoassay tubes for 3 h. After this period, the apex of the roots was removed from the solution, washed in running water for 30 s, and the varnish was scraped with a scalpel blade no. 13 to remove the radioisotope vestiges that might be present.

To evaluate the apical microleakage, the radioactivity emitted by the specimens was measured through a gamma camera (GE Millennium MG, Milwaukee, WI, USA) controlled by an acquisition computer (GenieAcq, GE, Milwaukee, WI, USA). For each tooth, a static image was acquired for two minutes at a 512 × 512 matrix size and zoom of 1.33. Regions of interest (ROIs) in each image were drawn over each tooth to obtain the total counts, maximums, and average using a proper software (Xeleris^TM^, GE, Milwaukee, WI, USA). The total counts obtained in each image were used to quantify the infiltration and thus the sealing ability.

### 2.7. Statistical Analysis

The statistical analysis was performed by ANOVA 1-factor with a Games–Howell post-hoc test, using IBM SPSS statistic software (version 27). The statistical significance was set at 0.05 (*p* < 0.05).

## 3. Results

The means and standard deviations (SD) of counts per minute (CPM) for the total of 2 min exposure of the specimens are given in Figure 3.

The highest infiltration of radioisotopes characterized by the highest values of CPM (counts per minute) in Figure 3 was observed in the positive group, followed by the TotalFill Fast Set Putty group. The MTA group and the negative group showed statistically less infiltration. The higher the infiltration of the radioisotope, the poorer the sealers’ sealing ability.

The key findings of this study are regarding the two experimental groups, where the TotalFill Fast Set Putty group (7335.8 ± 2755.5) showed a statistical increase in the total counts (*p* = 0.009) compared with the MTA group (4059.1 ± 1231.1).

Statistical analysis also showed a statistically significant increase (*p* < 0.001) of the total counts of the MTA group (4059.1 ± 1231.1) when compared with the negative group (202.2 ± 161.5). Regarding the positive group (12,746.0 ± 4497.7), the MTA group (4059.1 ± 1231.1) showed a significant decrease in the total counts (*p* = 0.039).

The TotalFill Fast Set Putty group (7335.8 ± 2755.5) showed a statistically significant increase in the total counts (*p* < 0.001) when compared with the negative group (202.2 ± 161.5). No significant difference (*p* = 0.168) was found between the TotalFill Fast Set Putty (7335.8 ± 2755.5) and the positive group (12,746.0 ± 4497.7).

## 4. Discussion

Immature permanent teeth may develop pulp necrosis due to trauma or caries with pulp involvement [2]. A suitable sealing ability is essential to the longevity of these teeth because it prevents bacterial leakage [5,6]. As it is known, the presence of microorganisms in the root canal is the main etiological factor of pulp diseases or endodontic unsuccess, so in order to obtain a suitable sealing ability, the choice of the root canal sealer is key [33]. Due to its physical, chemical, and biological properties, calcium silicate-based sealers seem to be an acceptable fit [5,23].

The present experimental study was designed to evaluate the apical sealing ability of two calcium silicate-based sealers, placed as an apical plug in simulated immature teeth, using a solution of sodium pertechnetate (^99m^TcNaO_4_). The radioactive decay of molybdenum originates the artificial element, the technetium, that decays with a half-life of 6 h by isometric transition and emission of 140.5 keV of gamma radiation [34].

Nevertheless, apical sealing ability can be studied through various methods such as the use of dyes, glucose penetration, bacterial or toxins infiltration, radioactive isotopes, protein microleakage, electrochemical microleakage, or scanning electron microscopy [35,36].

The most common methods applied to evaluate the sealing ability in recent studies are dye and bacterial leakage. Nonetheless, dye penetration using methylene blue has some limitations, such as the necessity to destroy the specimens, the subjective evaluation of the results that request an expertise operator, and the semiquantitative nature of measuring microleakage [37]. The bacterial infiltration method, although it might be more clinical and biologically relevant when compared to dye penetration, can only be applied in sealers that have an antimicrobial activity to the type of bacteria employed and usually have a qualitative rather than a quantitative nature [36]. The use of methods using radioactive isotopes such as ^99m^TcNaO_4_ has a quantitative and nondestructive nature supporting measurement of microleakage from the same specimens at intervals over extended periods, without the need to destroy the sample, contrasting with the two methods mentioned above [38]. It also allows quantitative detection of leakage per minute and can be quickly detected even when very small concentrations are present, which is an advantage over the dye penetration method. [39,40].

The substantially higher leakage present in the positive control group compared to the other groups in the current study shows that in the absence of an apical plug, the solution of ^99m^TcNaO_4_ was capable of penetrating [38]. On the other hand, the negative control group had significantly lower values of infiltration, which is an indication that two layers of varnish are sufficient to seal the root surfaces, preventing side infiltration and confirming the method used in this study [38].

There is some reluctance about the value of in vitro leakage studies in terms of clinical significance, limitations of the results, and lack of technique standardization. Based on available evidence in the study of Jafari et al., it seems that despite possible errors in leakage studies, they are particularly valuable [36]. Furthermore, sealing ability is responsible for preventing infiltration of bacteria and their byproducts into periapical tissue, as already stated [5]. Therefore, this characteristic remains a priority when evaluating and comparing new apical plug materials before they are used in the clinic [26,36].

The standardization of in vitro conditions is an important step for controlling possible biases and optimizing the statistical analysis [41]. In this sense, it was important in this research to try to standardize the internal anatomy of the sample [41] by using human extracted teeth with the inclusion criteria mentioned above. The restriction toward the characteristics of the sample performed a conditioning factor of the number of teeth included. The preparation of the sample was performed using a previously described retrograde instrumentation technique by Hachmeister et al. [4], with the modifications already described. Due to the importance of successful debridement of root canals during the instrumentation procedure, the irrigation process with chemical irrigants was firstly made with EDTA [39]. Still, using just EDTA is not sufficient to remove the entire smear layer, but 1 min irrigation of EDTA combined with a sodium hypochlorite solution has been reported to have the best results. [42]. A final irrigation step was made with saline since using the solutions such as EDTA and sodium hypochlorite together, without the saline, leaves crystals on the canal walls [42].

A thickness of 4 mm of the sealer was placed as an apical plug in this study since several studies have discovered that a 4 mm thickness of an apical plug material provides a significantly suitable seal [43,44,45]. In a previous study, Lertmalapong et al. concluded that TotalFill BC RRM Putty and ProRoot MTA at a thickness of 4 mm, exhibited the best sealing ability and marginal adaptation [5]. The possible reason given was that a greater mass of material results in larger expansion and reduction in gaps [46]. The calcium-silicate present in the composition of the sealers is what provides expansion of 0.2% to 0.6% of its initial volume, greatly contributing to its sealing capacity [46].

Materials based on tricalcium silicates such as WMTA and BC RRM are hydraulic because they are capable of setting in moisturized environments. [47]. As so, the obturated teeth were kept in floral foam to replicate in vivo periapical ligament moisture [48]. As the study of Caronna et al. shows, the bioceramic materials can absorb sufficient moisture from periapical fluids to set completely without the need of a moisturized cotton pellet [28].

The bioceramic root repair material used (BC RRM) was a fast set putty known in Europe as TotalFill by FKG, but in the USA, it is known as EndoSequence by Brasseler [49,50]. Consequently, Endosequence and TotalFill BC RRM putty have the same composition [26,27,32,51]. The premixed syringeable ready-to-use formulation provides the clinician with a homogeneous and consistent material [49,50,51]. Its fast-setting working time (20 to 30 min) is due to the significantly smaller particle size, which accelerates the setting reaction [32]. In addition, because the sealer is premixed with nonaqueous but water-miscible carriers, it will only set when exposed to an aqueous environment. Hence, it does not set when stored neither has the problem of heterogenous consistency during the manipulation mixture, as it might happen in handling WMTA [52,53].

There are few studies comparing the sealing ability of these two materials, and usually, it is used dye penetration of methylene blue or bacterial leakage of *Enterococcus faecalis.* To the best of our knowledge, there is no other study comparing the two materials using radioactive isotopes. Moreover, studies about the sealing ability of BC RRM compared to MTA have taken contradictory results [26,27,31,54,55].

Under the conditions of this study, there was a statistically significant difference between WMTA and TotalFill BCRRM Fast Set Putty (*p* = 0.009), where the BCRRM had the highest leakage. This was in agreement with a recent study on Endosequence BC RRM by Hirschberg et al. [55]. However, other studies reported no significant difference in the two materials [26,27,54] although, in the Nair et al. study, the BCRRM had more percentage of leakage [26]. On the other hand, Jeevani et al. showed that BC RRM had less leakage than MTA [54].

The discrepancies may be related to different methodologies. Matloff et al. did not report any correlation between dye penetration studies and radioisotope techniques [36]. Likewise, a study by Barthell et al. showed no correlation between dye penetration and bacterial leakage techniques [36]. The majority of studies have shown no correlation between the different methods [36].

The probable cause for significantly more leakage of TotalFill BC RRM Fast Set Putty compared to the WMTA in this study could be due to the fast-working time and putty consistency that was considered hard to introduce and manipulate into the root canal. Some affinity for the metal instruments used (MAP System and Schilder Pluggers) was also recorded. This might lead to a compromised marginal adaptation that preceded the poor sealing ability registered [5]. Another probable cause could be the presence of voids [31]. In the study of Alsubait et al., the biding deficit registered in BC RRM Fast Set Putty was mainly within the material itself [50].

The evaluation of apical infiltration after four days of placing the apical plug was chosen to allow the materials an adequate time for setting. Although setting time could be another factor for the results obtained, according to the manufacturer, the working time of BCRRM Fast Set Putty is around 30 min and setting time is nearly 4 h under normal conditions for BCRRM and MTA [26]. However, BCRRM might need up to 12 h under particularly dry conditions [26]. In the study of Guo et al., the initial and final setting time of ERRM Putty was significantly longer than WMTA, but on the 4th day of setting, it reached the same level of microhardness as WMTA [52]. For this reason, when the infiltration was performed on the 4th day, the two materials were in equal stages of setting.

The sealing ability of MTA has been widely investigated and documented in the literature. Within the results of our study, there was some apical infiltration of ^99m^TcNaO_4_ in the MTA group. Nevertheless, the values of infiltration were significantly lower than the positive control, thus corroborating the fact that MTA have suitable sealing ability. This is in agreement with previous studies that state that the incidence of leakage in MTA apical plug is related to the presence of failures within the sealer, at the sealer-dentin interface, and/or by the presence of pores [56].

Also, the presence of cracks or surface debris nonvisible through the microscope and alterations in the angulation of the root may highlight the appearance of gap defects [24] that led to the radioisotope infiltration in both experimental groups.

However, the results of in vitro studies cannot be immediately anticipated to the clinical conditions [27]. Although this study suggests that WMTA has better sealing ability than TotalFill BC RRM Fast Set Putty, more in vitro studies, with significantly larger sample size and measuring the infiltration at intervals over extended periods (at 7th and 28th days, for example) should be made for further investigation of these materials’ performance when subjected to in vivo conditions. Nevertheless, the search for an ideal apical plug material is still in pursuit. Until then, choosing the gold-standard MTA seems like a very thoughtful and literature-based decision.

## 5. Conclusions

More in vitro studies are required for additional support to the limited sample used and for additional periods of evaluation of the sealing ability. It also may be interesting to use this methodology with different sealers.

Within the limitations of this in vitro study, it can be concluded that for treatment of immature teeth requiring orthograde delivery of an apical plug, White ProRoot MTA has a sealing ability significantly better than TotalFill BC RRM Fast Set Putty.

There was a statistically significant difference in the sealing ability of the calcium silicate-based sealers White ProRoot MTA and the TotalFill BC RRM Fast Set Putty, thus rejecting the null hypothesis.

## Figures and Tables

**Figure 1 materials-14-06456-f001:**
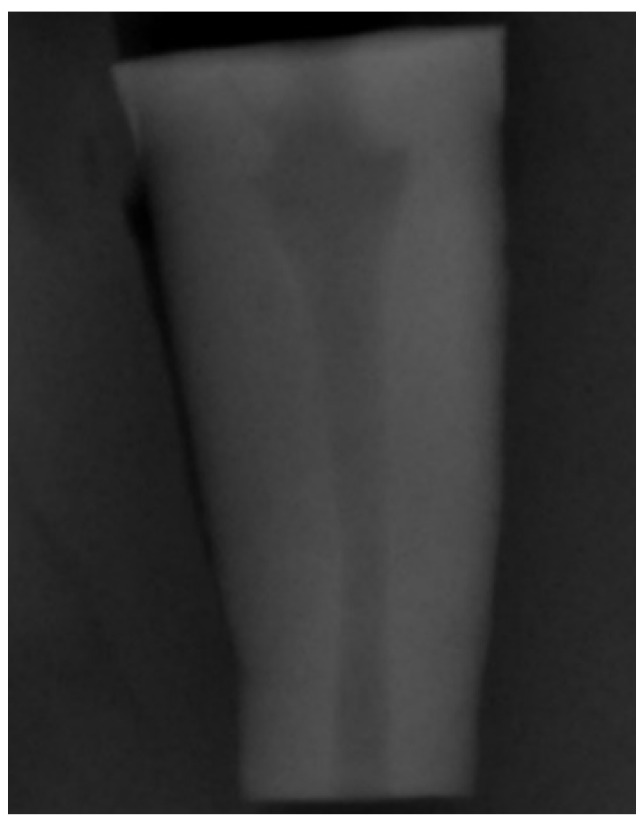
Periapical radiograph to demonstrate the simulation of the divergent apical architecture created with the retrograde instrumentation.

**Figure 2 materials-14-06456-f002:**
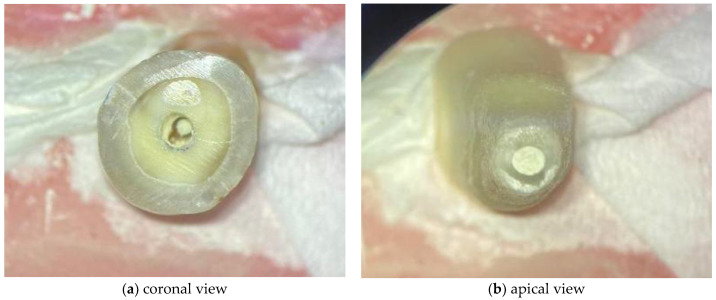
Images acquired under optical microscope for demonstrating the 4 mm apical plugs.

**Figure 3 materials-14-06456-f003:**
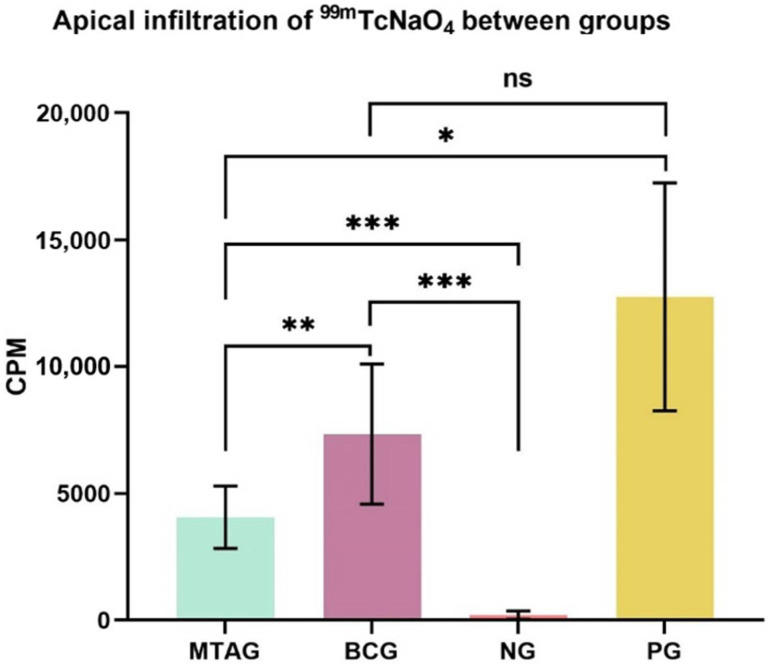
Visual representation of the mean values and standard deviation of the total counts in each group and level of significance between groups: * *p* < 0.05; ** *p* < 0.01; *** *p* < 0.001; ns—no significance. MTAG: MTA group; BCG: TotalFill Fast Set Putty group; NG: negative group; PG: positive group; CPM: counts per minute.

**Table 1 materials-14-06456-t001:** Manufacturer, composition, and lot no. of the endodontic cements used.

Material	Manufacturer	Composition	Lot No.
TotalFill BC RRM Fast Set Putty^®^	FKG, La Chaux-de-Fonds, Switzerland	Tricalcium silicate, dicalcium silicate, zirconium oxide, tantalum pentoxide, calcium sulfate (anhydrous) and calcium phosphate monobasic	2003FSPS
ProRoot WMTA^®^	Dentsply Maillefer, Ballaigues, Switzerland	Dicalcium silicate, tricalcium silicate, tricalcium aluminate, bismuth oxide, calcium sulfate, aluminum oxide, magnesium oxide, and iron oxide	0000172755

## Data Availability

The data presented in this study are available on request from the corresponding author.

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
