# Peer review of "Apical Sealing Ability of Two Calcium Silicate-Based Sealers Using a Radioactive Isotope Method: An In Vitro Apexification Model"

_materials, 2021, doi:10.3390/ma14216456_

Round 1

Reviewer 1 Report

This work aims to evaluate and compare the sealing ability of two calcium silicate based sealers based on isotope method. The research gave the evidence for the better performance on sealing ability of White ProRoot 413 MTA than TotalFill BC RRM Fast Set Putty. Although the author used the small samples with short time research, the results of this study has meaningful.

The visual observation of experimental process should be reported in the manuscript.

Additional comments:

This work aims to evaluate and compare the sealing ability of two calcium silicate-based sealers based on an isotope method. The research gave evidence for the better performance on sealing ability of White ProRoot 413 MTA than TotalFill BC RRM Fast Set Putty. Although the result of this study is meaningful. However, the research has some limitations and needs further studies. My comments are as follows:

  1. The amount of sample used in this study was small. 
  2. The sealing ability of materials was researched in quite short time.  
  3. The visual observation of the experimental process should be reported in the manuscript.
  4. The limitation of the research should be included in the conclusion and the further study of this research also needs to be given.

Author Response

We appreciate the time and work given to our manuscript. 

Our answers are below:

  1. The amount of sample used in this study was small. In our study, we highlight the limited sample and justified it due to the restrictions towards the characteristics of the sample that performed a conditioning factor of the number of teeth included. However, similar studies have been carried out with similar sample sizes.
  2. The sealing ability of materials was researched in quite short time.  According to the manufacturers' instructions, the two materials were at similar stages of setting to evaluate the sealing ability. However, we also state it as a limitation and suggest that more periods of time can be made in the future.
  3. The visual observation of the experimental process should be reported in the manuscript. We confirm that all the experimental work was performed and observed by the authors and supervised by an endodontic expert and we add this statement in the manuscript. 
  4. The limitation of the research should be included in the conclusion and the further study of this research also needs to be given. We took this comment into consideration and add this information in the conclusion section of the manuscript. 

Reviewer 2 Report

The research article presented by Pereira et al. focus on the evaluation and comparing the sealing ability of two calcium-silicate based cements. The authors did a reasonable work in comparing the sealing ability of two cement materials. Overall, the article was convincing and recommended for publication with minor revision. The results section was written too short and it can be improved by addressing the key findings.

Author Response

Thank you for your time and consideration.

We appreciate the comments given and we took them into consideration as we improved the results section present in the revised manuscript to better guide the reader to the meaning of the results and what are the main findings for the purpose of our study. 

Reviewer 3 Report

the study was prepared in a correct way. It should be noted that the topic has already been described, as emphasized by the authors.

However, it should be assumed that the study was fairly reliable and may be published, provided that some editorial corrections are made.

In my opinion, the description of the groups adopted and the trials carried out was incorrectly described. The reader does not know until the conclusion what each group means. Therefore, please re-edit this text so that from the very beginning of the analysis of the results it is known which groups and why were selected.

Author Response

We appreciate the time and work given to our manuscript. 

Editorial changes have been done.

Regarding the results section, we took the comments into consideration and contextualize the analysis from the beginning of this section to better guide the reader to the meaning of the results and what are the main findings for the purpose of our study. 

The description of the trials (endodontic sealers used) and also of the groups are present in the Materials and Methods and also in the subtitle of the figure 3 in the results section.